# Genome-Wide Identification of the *PIP5K* Gene Family in *Camellia sinensis* and Their Roles in Metabolic Regulation

**DOI:** 10.3390/genes15070932

**Published:** 2024-07-17

**Authors:** Xiaoping Wang, Yuanyuan Xiong, Xiaobo Tang, Ting Zhang, Weiwei Ma, Yun Wang, Chunhua Li

**Affiliations:** Tea Refining and Innovation Key Laboratory of Sichuan Province, Tea Research Institute, Sichuan Academy of Agricultural Sciences, Chengdu 610066, China; 18202817375@163.com (Y.X.); sctea@163.com (X.T.); 18780201475@163.com (T.Z.); shuilingxiao@126.com (W.M.); scteawl@163.com (Y.W.); sctea175@163.com (C.L.)

**Keywords:** tea, *PIP5K*, genome-wide, phylogenetic analysis, metabolic regulation, expression pattern

## Abstract

Spider mite infestation has a severe impact on tea growth and quality. In this study, we conducted a deep exploration of the functions and regulations of the *CsPIP5K* gene family using chromosomal localization and collinearity analysis. Additionally, we carefully examined the *cis* elements within these genes. To fully understand the metabolic response of *CsPIP5K* under spider mite infection, we integrated previously published metabolomic and transcriptomic data. Our analysis revealed that multiple *CsPIP5K* genes are associated with phospholipid metabolism, with *CsPIP5K06* showing the strongest correlation. Therefore, we employed qPCR and subcellular localization techniques to determine the expression pattern of this gene and its functional location within the cell. Overall, this study not only comprehensively elucidated the characteristics, structure, and evolution of the *CsPIP5K* gene family but also identified several candidate *CsPIP5K* genes related to phospholipid biosynthesis and associated with spider mites based on previously published data. This research makes a significant contribution to enhancing the resistance of tea to spider mite and maintaining optimal tea quality.

## 1. Introduction

Tea (*Camellia sinensis*), with a rich history spanning centuries in China, has traditionally been revered both as a refreshing beverage and a medicinal herb [1]. Today, it has evolved into one of the most sought-after drinks worldwide [2]. Its distinctive aroma and flavor are widely adored, solidifying its status as a staple in everyday life. However, tea is confronted with numerous challenges posed by various pests and diseases that can attack their roots, stems, leaves, flowers, and seeds, potentially leading to significant yield losses of up to 40% [3]. Despite these challenges, the enduring popularity and cultural significance of tea ensures that farmers and researchers will continue to strive for innovative solutions to protect this beloved beverage and its production.

The *PIP5K* gene family plays a vital regulatory role in plant cell signaling transduction [4,5,6]. The enzymes encoded by this family catalyze the conversion of phosphatidylinositol 4-phosphate (PI4P) to phosphatidylinositol 4,5-bisphosphate (PI(4,5)P2), thereby occupying a central position in plant cell signaling processes [7,8]. Kinase domains and specific substrate binding sites, conserved among family members, facilitate the precise control of PI(4,5)P2 synthesis and distribution [9]. *PIP5K* performs a crucial enzymatic function in the inositol signal transduction system. The gene family encodes multiple isoenzymes of *PIP5K*, which are abundant in higher plants’ genomes compared to animals and fungi [10]. Specifically, the *Arabidopsis* genome contains 11 *PIP5K* genes that share a significant homology with type I PtdInsP kinases (*PIP5K*s) found in animals [4]. In summary, the *PIP5K* gene family is crucial in plant cell signaling, regulating PI(4,5)P2 synthesis. In recent years, the diverse functions of *PIP5K* in plant growth and development have been progressively uncovered [2,11,12]. Numerous studies have delved into the role and mechanisms of *PIP5K* across various plant species [13,14,15]. For example, in *Arabidopsis thaliana*, *PIP5K* has demonstrated its ability to regulate root hair tip growth, promoting root hair elongation through its influence on cytoskeletal organization [16]. Furthermore, leveraging advances in transcriptome and metabolomics technologies, a growing number of studies are examining the expression patterns and regulatory networks of *PIP5K* in specific crops [17,18]. These investigations are significantly advancing our understanding of the diverse functions of *PIP5K* and establishing a solid theoretical foundation for future explorations into its potential applications in tea plant growth regulation.

Since *PIP5K* represents an important branch in the phosphatidylinositol (PI) cycle, most studies have focused on its roles in plant growth and development, while neglecting its ability to resist biotic and abiotic stresses. In the context of stress response, *PIP5K* has proven responsive to challenging environmental conditions, such as drought and salt stress, bolstering plant resilience by modulating intracellular signal transduction pathways [19,20,21]. However, when exposed to stress conditions induced by mannitol or NaCl, the *pip5k7*, *pip5k8*, and *pip5k9* triple mutant exhibited a greater suppression of main root growth and reduction in proximal root meristem size compared to the wild type [16]. The transcriptomic profiling of two soybean cultivars, drought-tolerant *SS2-2* and drought-sensitive *Taekwang*, was performed under normal and drought conditions, revealing that the *PIP5K* gene may be related to drought tolerance [11]. In biological stress, studies have revealed that through the application of cell permeabilization and short-term nonequilibrium labeling, the expression of *AtP5K1* in Baculovirus-infected insect (*Spodoptera frugiperda*) cells directly leads to the synthesis of PtdIns(4,5)P2 and PtdIns(3,4,5)P3 [22]. Previous studies have indicated the pivotal role of *PIP5K* in both biological and abiotic stresses. However, compared to other gene families such as the *NAC*, *MYB*, and *WRKY* family, there are still many unknowns about *PIP5K*, especially its behavior under spider mite infestation.

*Camellia sinensis*, as a significant economic crop, has been extensively studied for its growth patterns, developmental stages, and stress responses. The most detrimental and stubborn pests affecting tea production are spider mites, which can severely damage tea plants and reduce yield and quality [23,24]. Current research on the *PIP5K* gene family’s role in *Camellia sinensis*, particularly in relation to spider mite, remains limited, necessitating further exploration. Thus, we undertook a thorough investigation of the *CsPIP5K* gene family in tea plants, elucidating their distribution, phylogenetic relationships, and regulatory mechanisms. Through the integration of metabolomics and transcriptomics based on previously published data under spider mite infestation, we delved into the metabolic function of *CsPIP5K* genes. Finally, we excavated a gene named *CsPIP5K06*, which has the possibility of regulating a variety of lipids and is associated with spider mite infestation. We elaborated on various properties of this gene by qPCR and subcellular localization. These findings improve our understanding of the phosphatidylinositol-4-phosphate 5-kinase gene family’s role in tea plants. They form a crucial basis for refining tea cultivation, breeding, and stress resistance research, fostering sustainable tea industry advancement.

## 2. Materials and Methods

### 2.1. Identification of the Tea CsPIP5K Family

To identify potential *PIP5K* family genes in tea, the reference *Camellia sinensis* genome and annotation files were downloaded from the BIG database (accession no. GWHACFB00000000; https://bigd.big.ac.cn/search/?dbId=gwh&q=GWHACFB00000000, accessed on 4 August 2020) [25]. Additionally, the Ensembl Plants database (http://plants.ensembl.org/, accessed on 10 March 2024) was consulted to acquire genome data for *Arabidopsis*, tomato, and grape, allowing for comparative analysis. Subsequently, the hidden Markov model (HMM) for the PIP5K domain (ID: PF01504) was retrieved from the Pfam database (http://pfam-legacy.xfam.org/, accessed on 10 March 2024). This HMM was then employed using the HMMER tool to search and compare the PIP5K domain across the entire genome of tomato proteins, with an E-value threshold of ≤1 × 10^−5^. Lastly, all identified proteins were verified through the NCBI Conserved Domain Database program (https://www.ncbi.nlm.nih.gov/, accessed on 10 March 2024) and the Pfam database, utilizing the search functionality of online software to ensure the accuracy of our findings. Utilizing the ExPASy (https://web.expasy.org, accessed on 11 March 2024) and Cell-PLoc 2.0 (http://www.csbio.sjtu.edu.cn/bioinf/Cell-PLoc-2/, accessed on 11 March 2024) online software, we determined the physicochemical properties of the tea PIP5K protein [26].

### 2.2. Structural Analysis of the CsPIP5K Gene Family in Tea

Using the MEGA11 software (version 11.0.10), we compared the PIP5K protein sequences of tea, *Arabidopsis* (*Arabidopsis thaliana*, *At*), tomato (*Solanum lycopersicum*, *Sl*), and grape (*Vitis vinifera*, *Vv*), employing the Muscle algorithm [27]. Subsequently, the neighbor joining (NJ) method was applied to analyze and construct a phylogenetic tree for the protein family. A bootstrap repeat value of 1000 was chosen to ensure robustness in the tree construction. To visually enhance the clarity and presentation of the phylogenetic tree, we utilized the online platform iTOL (https://itol.embl.de/) [28]. Additionally, conservative motif analysis was performed using the MEME tool (Version 5.5.5, https://meme-suite.org/tools/meme, accessed on 4 May 2024) [29]. Finally, TBtools software (version 2.030) was employed to generate gene structure maps and protein domain distribution maps, providing a comprehensive visualization of the genetic and proteomic features [30].

### 2.3. Chromosome Location and Collinearity Analysis of the CsPIP5K

Utilizing the chromosome position information of the *PIP5K* genes, a position distribution map was crafted with TBtools software [30]. Furthermore, MCScanX (version 1.0) was employed to conduct a collinearity analysis of the *PIP5K* genes across tea, *Arabidopsis*, tomato, and grape [31]. Additionally, the Circos tool (version 0.69-9) was leveraged to generate a chromosome collinearity map, providing a visual representation of the gene arrangement and relationships.

### 2.4. Cis Element Analysis of the CsPIP5K Gene in Tea

Employing TBtools software, we retrieved the 2000 bp upstream sequence flanking the *CsPIP5K* genes. These sequences were then subjected to in-depth analysis using the Plant Care online platform (http://bioinformatics.psb.ugent.be/webtools/plantcare/html/, accessed on 4 May 2024), which allowed us to precisely identify and visualize the cis-acting elements within the predicted promoter regions [32].

### 2.5. Analysis of the Expression Pattern of CsPIP5Ks under the Spider Mite Infestation

Using the transcriptome data already published by our laboratory [33], we analyzed the expression pattern of the *CsPIP5K* genes in different varieties were induced by spider mite. These data correspond to the accession number PRJNA996324. Subsequently, we employed the “pheatmap” package (version 1.0.12) in R language to generate heatmaps depicting the expression patterns of *CsPIP5K* genes.

### 2.6. Integrated Analysis of Metabolomics and Transcriptomics

Based on the previously published metabolic profiling data of different tea varieties infected with spider mite at various time points from our laboratory, in combination with the corresponding transcriptomic profiles, we utilized the cor.test() function in R language to calculate the Pearson correlation coefficients between transcription level and metabolite abundance of *CsPIP5K* gene family members. Since *PIP5K* is closely associated with phospholipid metabolism, we extracted phospholipid-related metabolites from the overall metabolic profiling data. Screening was conducted using a threshold of an absolute correlation coefficient greater than 0.8 and a significant *p*-value less than 0.01.

### 2.7. RNA Extraction

Total RNA was extracted from multiple tissues of “*Tianfu-5*” (a widely studied tea variety, abbreviated as TF5) using Vazyme (RC411) Kit according to the manufacturer’s specifications. The yield of RNA was determined using a NanoDrop 2000 spectrophotometer (Thermo Scientific, Waltham, MA, USA), and the integrity was evaluated using agarose gel electrophoresis stained with ethidium bromide.

### 2.8. Real-Time Quantitative RT-PCR

Quantification was performed with a two-step reaction process: reverse transcription (RT) and PCR. The step is 1.0 μg RNA, 10 μL of 2×TS Fly Reaction Mix, 1 μL of TransScript Fly RT/RI Enzyme Mix, 1 μL of Anchored Oligo (dT), add Nuclease-free H_2_O to 20 μL. Reactions were performed in a T100 Thermal Cycler (BIO-RAD, Hercules, CA, USA) for 5 min at 42 °C, 5 s at 85 °C. Real-time PCR was performed using LightCycler^®^ 480 II Real-time PCR Instrument (Roche, Basel, Switzerland) with 10 μL PCR reaction mixture that included 1 μL of cDNA, 5 μL of 2 × ChamQ Universal SYBR qPCR Master Mix (Vazyme, Nanjing, China), 0.5 μL of forward primer, 0.5 μL of reverse primer and 3.0 μL of nuclease-free water. Reactions were incubated in a 96-well optical plate (Roche, Swiss) at 95 °C for 5 min, followed by 40 cycles of 95 °C for 10 s, 60 °C for 30 s. Each sample was run in triplicate for analysis. At the end of the PCR cycles, melting curve analysis was performed to validate the specific generation of the expected PCR product. The primer sequences were designed in the laboratory and synthesized by Sangon Biotech (Shanghai, China) based on the mRNA sequences obtained from the NCBI database (Appendix A). The expression levels of mRNAs were normalized to GAPDH and were calculated using the 2^−ΔΔCt^ method [34].

### 2.9. Subcellular Localization of CsPIP5K

The subcellular localization of *CsPIP5K* was investigated using the tobacco transient expression system. Initially, the gene was cloned and inserted into the pR101-GFP vector. Both the constructed vector and the empty vector were then transformed into Agrobacterium tumefaciens GV3101. After verification, selected positive clones were cultured overnight and subsequently expanded at a 1:100 ratio. The bacteria were collected, resuspended in injection buffer to an OD600 of 0.6, and injected into healthy tobacco leaves. Following a 48–72 h incubation, observations and photographs were taken for initial localization. To further validate the results, co-transformation with the corresponding marker was performed for colocalization studies. Through these steps, the precise subcellular localization of *CsPIP5K* was determined.

## 3. Results

### 3.1. Characterization of Members of the CsPIP5K Gene Family in Camellia sinensis

In this study, a genome-wide identification analysis was conducted in *Camellia sinensis*, leading to the initial discovery of 17 putative *CsPIP5K* genes. These genes were subsequently named *CsPIP5K01* to *CsPIP5K17*, based on their chromosomal positions within the genome. A detailed analysis of the physicochemical properties of these proteins revealed that the amino acid residue counts within the *CsPIP5K* family varied significantly, ranging from a minimum of 122 aa in *CsPIP5K01* to a maximum of 1847 aa in *CsPIP5K17*. Similarly, protein molecular weights spanned a broad range, with CsPIP5K01 exhibiting the lowest weight of 13,713 Da and *CsPIP5K17* the highest at 20,4618 Da. Isoelectric points of the proteins also displayed considerable diversity, with *CsPIP5K01* possessing the lowest value of 4.43 and *CsPIP5K04* the highest at 9.02 (Appendix A).

### 3.2. Chromosomal Localization of the CsPIP5K Gene Gamily

The analysis of chromosomal localization revealed that the 17 members of the tea *CsPIP5K* gene family were scattered across 15 chromosomes (Figure 1). Notably, Chr01 (2 genes), Chr02 (2 genes), Chr04 (3 genes), Chr06 (3 genes), Chr10 (2 genes), and Chr13 (2 genes) were the primary loci for the *CsPIP5K* genes, while Chr05, Chr07, Chr08, Chr12 and Chr14 were devoid of any members from this gene family. The remaining three chromosomes each harbored a single *PIP5K* member. The distribution pattern of *CsPIP5K* genes within these chromosomes exhibited heterogeneity. Most of the *CsPIP5K* genes are located relatively far from each other on the same chromosome, and they are mainly found in the middle to lower parts of the chromosome. However, the *CsPIP5K* genes on Chr03, Chr13, and Chr15 exhibit a distribution pattern at both ends of the chromosome.

### 3.3. Phylogenetic Classification of the CsPIP5K Gene

To elucidate the phylogenetic relationship among *PIP5K* genes, a phylogenetic tree was constructed based on the protein sequences of 56 conserved structural domains from *PIP5K* genes in four species: *Arabidopsis thaliana*, tomato (*Solanum lycopersicum*), grape (*Vitis vinifera*), and tea. This phylogenetic analysis categorized the *PIP5K* genes into four distinct groups, designated as I, II, III, and IV (Figure 2). Genes within the same group exhibited closer phylogenetic relationships.

Specifically, group I comprised 10 *PIP5K* genes, including 4 from tea, 2 from *Arabidopsis*, 2 from tomato, and 2 from grape. Group II contained 10 *PIP5K* genes, distributed as 4 from tea, 4 from *Arabidopsis*, 1 from tomato, and 1 from grape. Group III encompassed 12 *PIP5K* genes, with 4 from tea, 3 from *Arabidopsis*, 3 from tomato, and 2 from grape. Group IV contained 24 *PIP5K* genes, distributed as 5 from tea, 8 from *Arabidopsis*, 7 from tomato, and 4 from grape.

Genes belonging to the same group often share similar functional characteristics. Notably, the majority of tea *CsPIP5K*s fell within group IV, indicating that these genes primarily perform specific functions within tea plants. This phylogenetic analysis provides insights into the evolutionary relationships and potential functional roles of *PIP5K* genes across these species.

### 3.4. CsPIP5K Gene Family Member Motifs and Gene Structures

As depicted in Figure 3, the PIP5K proteins exhibit a diverse array of motif types. An analysis of gene structure reveals that the tea *CsPIP5K* genome varies in the number of introns, ranging from two introns in *CsPIP5K01* to fourteen introns in *CsPIP5K07* and *CsPIP5K11*. The majority of *CsPIP5K* genes contain seven to eleven introns. Introns are sequences that interrupt the linear expression of genes, thus enhancing gene length, promoting gene recombination, favoring species evolution, and exerting regulatory effects [10,35]. These results suggest that *CsPIP5K* genes may play a positive role in tea evolution and have widespread regulatory functions. In particular, *CsPIP5K07* and *CsPIP5K11* may be more beneficial for the evolution of tea. When combined with the analysis of the evolutionary tree, it becomes evident that members of the same gene family group exhibit strong similarity (Figure 3). For instance, *CsPIP5K03*, *CsPIP5K02*, *CsPIP5K10*, and *CsPIP5K15* belong to the same group as *CsPIP5K12*, displaying comparable relationships. Additionally, the conserved motifs indicate that these five *CsPIP5K* genes share similar gene structures, with highly homologous motifs. Evidently, members within the same group demonstrate closer clustering relationships and comparable gene structures, indicating functional similarity. Furthermore, a search for ten motifs among 17 tea PIP5K proteins revealed that three subfamilies of 11 PIP5K proteins share motifs 2, 7, 4 and 10. This suggests that these motifs are widely distributed and strongly conserved within the tomato PIP5K protein sequence. Notably, most *CsPIP5K* members within the same subfamily also exhibit conserved motifs, indicating functional conservation. Notably, *CsPIP5K01* and *CsPIP5K05* contain motif 4 and motif 2 exclusively, respectively, potentially indicating a functional loss in these specific members within the *CsPIP5K* family.

### 3.5. Analysis of Protein Structural Domains of Tea CsPIP5K Gene Family Members

The distribution of structural domains within the CsPIP5K protein family is presented in Figure 4. Our analysis shows that all members of the *CsPIP5K* family share a common denominator: the PIP5K structural domain. Furthermore, one of these proteins sports an additional Phi1 superfamily structural domain at its C-terminus. Proteins belonging to the Phi1 superfamily participate in various cellular activities, such as cell proliferation, differentiation, apoptosis, and stress responses. Although these functions may also be present in tea plants, further research is needed to verify this hypothesis.

Among the 17 CsPIP5K proteins studied, four contain the Fab1_TCP domain, while three specific proteins—CsPIP5K05, CsPIP5K11, and CsPIP5K17—harbor the FYVE domain. The remaining proteins solely possess the PIP5K structural domain. Notably, CsPIP5K05 demonstrates the greatest diversity of domains, indicating its potential for expressing multiple functions simultaneously in tea plants.

### 3.6. Analysis of Covariance and Evolutionary Pressure Analysis for Members of the CsPIP5Ks

In the present study, we have delved into the intricate relationships within the tea *CsPIP5K* gene family by constructing a circos map (Figure 5). Our synteny analysis of 17 members of the *CsPIP5K* gene family revealed four homologous pairs exhibiting covariance (Appendix A). Notably, all these pairs undergo segment duplication, indicating the presence of gene duplication within the *CsPIP5K* gene family. This observation suggests that the *PIP5K* gene may have undergone an expansion of its family members through duplication events during evolution.

To delve deeper into the evolutionary ties among the members of the *PIP5K* gene family, this study constructed a synteny map encompassing the *Arabidopsis*, tomato, grape, and tea families (Figure 6 and Appendix A). The findings revealed the existence of direct homologous *PIP5K* gene pairs across *Arabidopsis thaliana*, tomato, grape, and tea. Notably, the number of homologous gene pairs between tea and tomato exceeded those between tea and *Arabidopsis thaliana*, as well as between tea and grape. This observation suggests a closer homologous evolutionary relationship between the *PIP5K* gene families of tea and tomato.

### 3.7. Analysis of Cis-Acting Elements of CsPIP5K Promoter

The *CsPIP5K* family was analyzed for *cis*-acting elements, categorized as light responsive, stress responsive, and phytohormone responsive (Figure 7A). Among these *CsPIP5K* genes, light response elements accounted for the largest proportion, reaching 46.28%, including elements such as GT1-motif, ACGT-containing element (ACE), and MYB-recognition element (MRE). Notably, *CsPIP5K16* and *CsPIP5K05* were the genes with the highest enrichment of these elements. Among the *CsPIP5K* genes, the primary element for drought stress was MBS, which was distributed across seven genes, while low-temperature responsiveness was primarily represented by LTR, found in three genes. Additionally, plant growth and developmental elements were identified, with 2 GCN4_motifs, indicating a role in endosperm expression, differentiation, organ regeneration, and tea growth/development regulation. Other elements were linked to circadian rhythms with the sequence CAAAGATATC, an element involved in differentiation of the palisade mesophyll cells (HD-Zip-motif).

Five of these *cis*-acting elements are related to phytohormone responses, including auxin response elements (AuxRR-core), gibberellin response elements (GARE-motif and P-box), abscisic acid response elements (ABRE), methyl jasmonate response elements (TGACG-motif), and salicylic acid response elements (TCA-element). Abscisic acid and salicylic acid are effective in controlling plant stress resistance, with 24 and 7 response elements respectively. Methyl jasmonate can induce a defense response in plants, thus demonstrating strong resistance to diseases, with the highest number of response elements totaling [36,37]. Phytohormones such as auxin and gibberellin can assist in plant growth [38,39]. Therefore, this suggests that multiple hormones may be involved in regulating the expression of *PIP5K*, jointly promoting tea plant stress tolerance and growth and development processes.

### 3.8. Analysis of Expression Pattern of CsPIP5K under the Spider Mite Infestation

To further investigate the biological functions of *CsPIP5K* genes under biological stress, we conducted a detailed analysis of the expression patterns of *CsPIP5K* genes in two tea varieties (FD and TF5) under spider mite infestation using our lab RNA-seq data [33]. As shown in Figure 8A, the expression profile clearly revealed that the expression levels of most *CsPIP5K* genes were generally low at the initial stage of infestation (2 days), while *CsPIP5K08* exhibited a relatively high expression level (Appendix A). However, as the treatment duration progressed, significant changes in the expression levels of these genes were observed. Notably, the expression levels of *CsPIP5K05*, *CsPIP5K06*, *CsPIP5K07*, and *CsPIP5K17* peaked in both tea varieties after 8 days of treatment, indicating it was strongly induced by spider mite. In contrast, the expression levels of *CsPIP5K09* and *CsPIP5K13* remained low throughout the 8-day treatment period. These findings reveal specific expression patterns of *CsPIP5K* genes in tea plants was induced by spider mite, further suggesting functional differences among different members of the gene family.

### 3.9. Metabolic Regulation of CsPIP5K Family Genes under Spider Mite Infection

In the present study, we conducted a comprehensive analysis of the transcriptional and metabolic profiles of the *CsPIP5K* family genes in relation to spider mite infestation. Given the pivotal role of PIP5K enzymes in phospholipid metabolism, we specifically focused on correlating the expression patterns of these genes with phospholipid-related metabolites.

Our correlation analysis revealed a total of 76 significant relationships between the *CsPIP5K* genes and phospholipid metabolites, including Phosphatidyl choline (PC), Phosphatidylinositol (PI), Phosphatidic acid (PA), Phosphatidyl ethanolamine (PE) and Phosphatidylglycerol (PG). Of these, 16 relationships were positive, while 60 were negative (Appendix A). This finding suggests a complex regulatory network involving the *CsPIP5K* family genes and their modulation of phospholipid metabolism during spider mite infection. Notably, five genes—*CsPIP5K05*, *CsPIP5K06*, *CsPIP5K13*, *CsPIP5K14*, and *CsPIP5K17*—stood out as having a particularly high number of significant correlations with phospholipid metabolites (Figure 8B). Among these, *CsPIP5K06* exhibited the most robust association, with 22 significant relationships, predominantly negative (90% of the total). This observation indicates that *CsPIP5K06* may play a central role in the downregulation of phospholipid metabolism during spider mite infection.

To further validate our findings, we examined the expression patterns of these genes in two cultivars infected with spider mite for eight days. Consistent with our correlation analysis, the expression level of *CsPIP5K06* was significantly upregulated in both cultivars (Figure 8A). This upregulation suggests that *CsPIP5K06* is induced by spider mite infection and functions primarily to repress phospholipid metabolism.

### 3.10. Analysis of Multi-Tissue Expression Profile and Subcellular Localization of CsPIP5K06

Given that *CsPIP5K06* may play a crucial role in lipid metabolism, for example, we followed up with a more detailed analysis of this gene. We preferred to identify the expression profile of this gene in several tissues of tea tree, selected from T-1 (first leaf of new tip), T-2 (2nd leaf of new tip), T-3 (3rd leaf of new tip), T-Y (bud of a new tip), CY (mature leaf), NJ1 (tender stem), MJ (ramie stalk), XG (fibrous root), ZG (taproot) tissues, a total of six tissues, and each biological replicate contains three technical replicates. From the results, the three biological replicates all have small error bar, indicating that the experimental results have good accuracy and consistency. Moreover, they all demonstrated that the gene was highly expressed at CY, followed by T and MJ, and the expression level at CY was much higher than that at other tissues (Figure 9A). Spider mite will mainly parasitize the mature leaves of tea and damage the leaf tips. Based on our RNA-seq results showing that this gene is also induced expression by spider mite (Figure 8A), so this result flanks the validation that *CsPIP5K06* may be associated with spider mite.

Using the tobacco transient expression system, we conducted an in-depth investigation into the subcellular localization of the gene *CsPIP5K06*. Preliminary localization experiments revealed that the gene was localized in the mitochondria (Figure 9B). To further validate this finding, we employed colocalization technology, co-transforming the target gene with the red mitochondrial marker vector mCherry and repeating the colocalization experiment. The results indicated that *CsPIP5K06* was indeed localized in the mitochondria, contradicting the subcellular prediction outcomes. This discovery provides crucial insights into the function and regulatory mechanisms of this gene within the cell.

## 4. Discussion

This comprehensive study delved deeply into the metabolic regulatory function of the *CsPIP5K* gene family in tea plants during red spider infestation. It uncovered the critical role of this gene family in phospholipid metabolism and its potential impact on the defense mechanism of tea plants against spider mite. These findings are consistent with previous studies that pointed out the involvement of phospholipid metabolism in plant defense responses, especially in the production of defensive compounds and signaling molecules [40,41,42,43,44]. Through meticulous chromosomal localization and phylogenetic analysis, we determined the precise location of the *CsPIP5K* gene family in the tea plant genome and traced its evolutionary lineage (Figure 1 and Figure 2). Our findings align with previous genetic mapping studies in other plant species [17,45], which suggest that genes involved in phospholipid metabolism tend to be unevenly distributed across chromosomes.

Our results contribute to a deeper understanding of the complex metabolic networks behind plant defense mechanisms. Previous studies have shown that plants respond to biological stresses, such as pest infestation, through a series of biochemical and physiological changes involving metabolic reprogramming [46,47]. By integrating transcriptome and metabolome datasets, we elucidated the precise metabolic regulatory functions of the *CsPIP5K06* gene family during spider mite infestation. Our findings align with recent studies that emphasize the importance of integrating multi-omics approaches for a comprehensive understanding of plant responses to biological stresses. Specifically, we identified a significant correlation between the *CsPIP5K06* gene and specific phospholipid metabolites (Figure 8B), suggesting a potential role for this gene in coordinating the defense mechanism of tea plants against spider mite.

Regarding the specific mechanism of *CsPIP5K06*, although this study could not delve into the details, our work was guided by several clues. Promoter analysis revealed the presence of numerous light responsiveness elements in the gene’s promoter region, along with phytohormone responsive elements (Figure 7B). These findings suggest that the gene may be regulated by transcription factors such as *HY5*, *WRKY*, and *AP2/ERF*, which have been reported extensively in plant defense against biological stresses. For instance, Chen et al. discovered that *HY5*, a key factor in plant phototransduction, plays a positive regulatory role in plant resistance to downy mildew [48]. Studies have shown that *ARF8* can directly bind to the promoter of the key disease resistance regulatory gene *WRKY45*, inhibiting its transcription and thereby regulating disease resistance [49]. In rice, infection by *Magnaporthe grisea* can induce the expression of *ERF* genes such as *OsBIERF1*, *OsBIERF3*, *OsBIERF4*, and *OsEBP2* [50]. Additionally, we found that multiple *CsPIP5K* genes were highly expressed in FD-8d samples, while their expression levels decreased in the TF5 variety during the same period (Figure 8A). However, *CsPIP5K05*, *06*, and *17* maintained relatively high expression levels. We hypothesize that other genes may also play a role in response to spider mite, and these three genes may have a broader spectrum of activity in tea plants.

Collectively, our results provide novel insights into the metabolic regulation of the *CsPIP5K* family genes during spider mite infection. Specifically, we identify *CsPIP5K06* as a key player in modulating phospholipid metabolism in response to this stress condition. These findings have potential implications for understanding the molecular mechanisms underlying plant defense and may pave the way for the development of novel strategies for pest management.

## 5. Conclusions

In summary, our study provides insights into the role of the *CsPIP5K* gene family during spider mite infestation in *Camellia sinensis*. This study, for the first time, identified 17 PIPK family genes in *Camellia sinensis* and classified them into four clusters. It also elucidated the relationship between these genes’ involvement in duplication events and species evolution. Additionally, using previously published multi-omics data, we identified that *CsPIP5K* is associated with spider mite infestation by regulating phospholipids at the mitochondria. Overall, this research will enhance our understanding of the CsPIP5K gene family, paving the way for improved tea cultivation and pest management.

## Figures and Tables

**Figure 1 genes-15-00932-f001:**
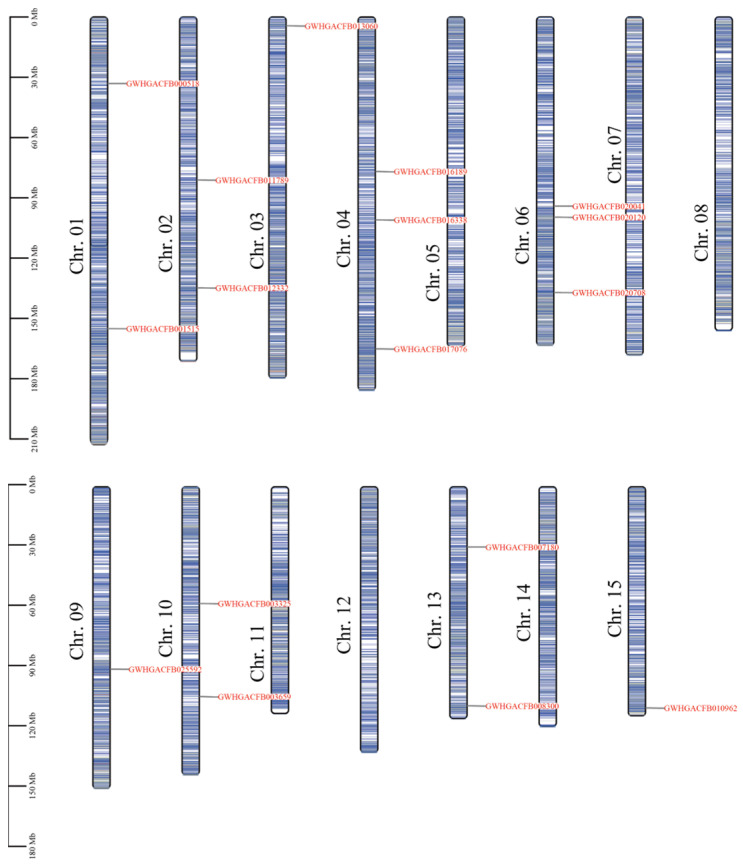
The chromosomal localization of the tea *CsPIP5K* gene family. Chromosome names are listed on the left, with gene locations indicated by black lines. The color represents gene density, ranging from low (white) to high (bule).

**Figure 2 genes-15-00932-f002:**
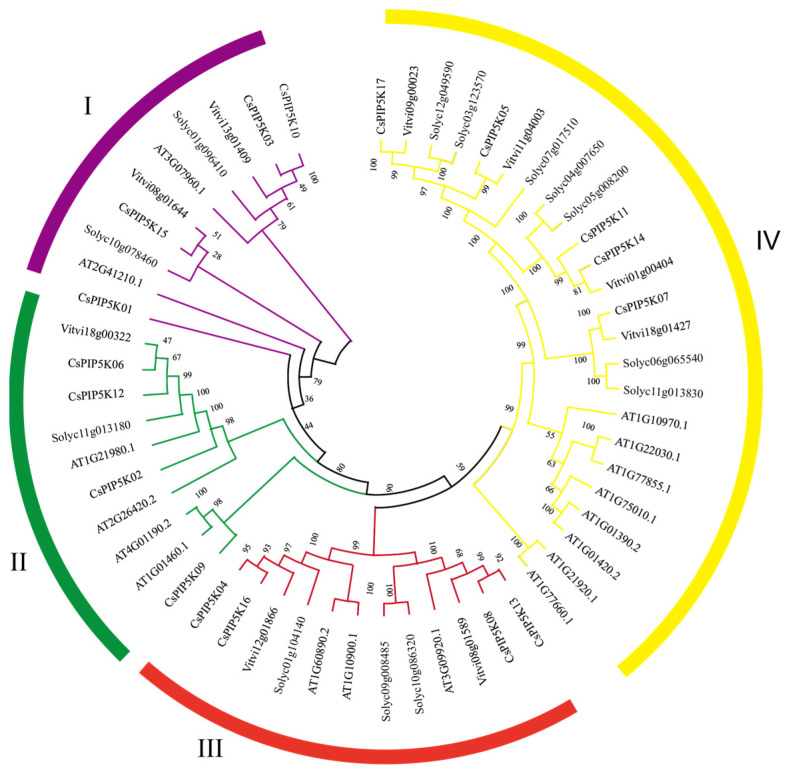
A phylogenic tree depicts *PIP5K* gene family members from *Camellia sinensis* (*Cs*), *Arabidopsis thaliana*, *Solanum lycopersicum*, and *Vitis vinifera*. Each clade is distinguished by a unique color.

**Figure 3 genes-15-00932-f003:**
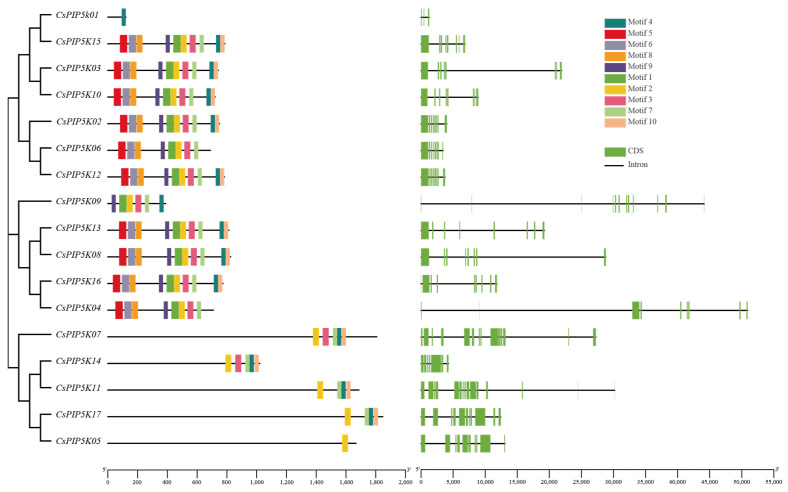
Phylogenetic, gene structure, and motif analysis of *CsPIP5K*. The picture on the left shows a phylogenetic tree of 17 *CsPIP5K* proteins. The unrooted neighbor-joining phylogenetic tree was constructed with MEGA11 using full-length amino acid sequences of 17 CsPIP5K proteins, and the bootstrap test replicate was set as 1000 times. The middle picture, the distributions of conserved motifs in *CsPIP5K* genes. Ten putative motifs are indicated in different colored boxes. On the right, exon/intron organization of *CsPIP5K* genes. Green boxes represent exons and black lines with same length represent introns.

**Figure 4 genes-15-00932-f004:**
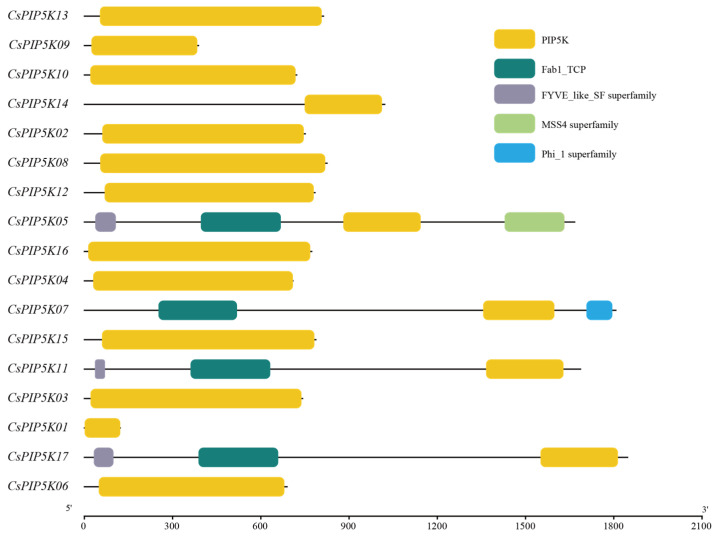
Protein domain distribution of CsPIP5K family members in tea. The length of different lines reflects the length of genes, and boxes of different colors on the lines represent domains with different information.

**Figure 5 genes-15-00932-f005:**
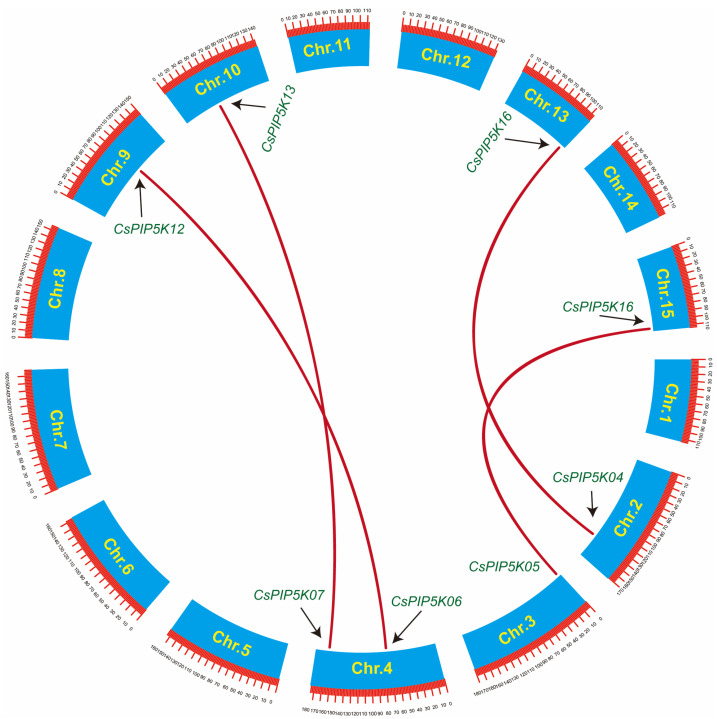
A synteny analysis of *CsPIP5K* family genes in tea. Different blocks represent different chromosomes of tea plants, and red lines specifically highlight syntenic gene pairs of *CsPIP5K*.

**Figure 6 genes-15-00932-f006:**
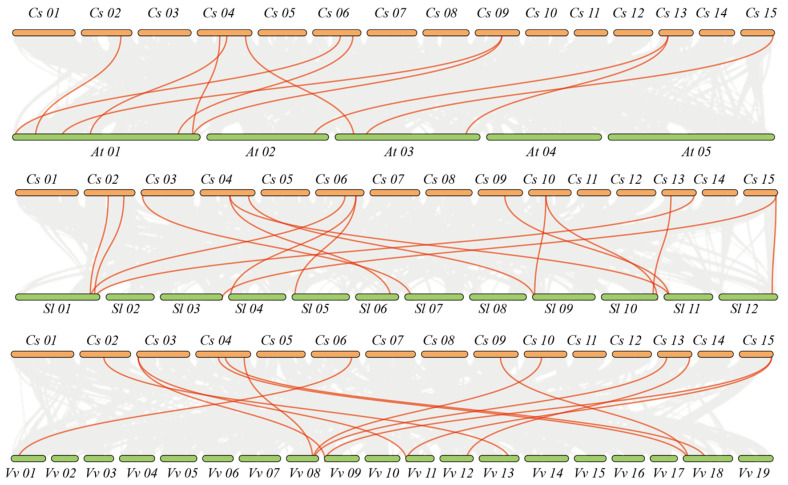
Interspecific synteny relationship between *CsPIP5K* gene family members and *Arabidopsis thaliana* (*At*), *Solanum lycopersicum* (*Sl*), and *Vitis vinifera* (*Vv*). Gray lines represent collinear blocks between tea and other plant genomes, with red lines emphasizing syntenic gene pairs of *CsPIP5K*.

**Figure 7 genes-15-00932-f007:**
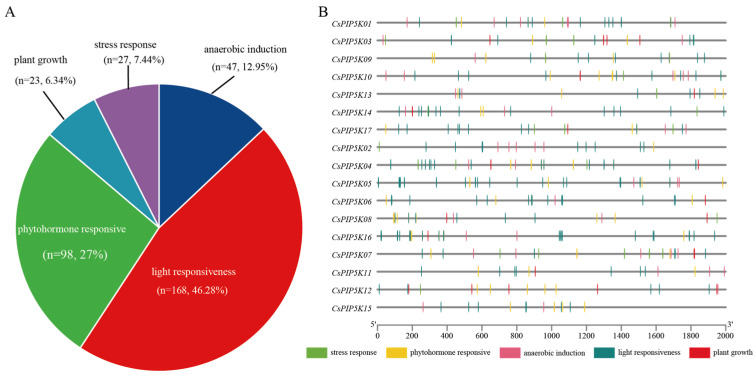
Analysis of the *CsPIP5K* family promoters. (**A**) Pie chart summarizing the functional elements; (**B**) distribution of promoter elements within the *CsPIP5K* family. The gray line represents the 2000 sequence of the gene promoter region, and boxes of different colors on the line represent different elements.

**Figure 8 genes-15-00932-f008:**
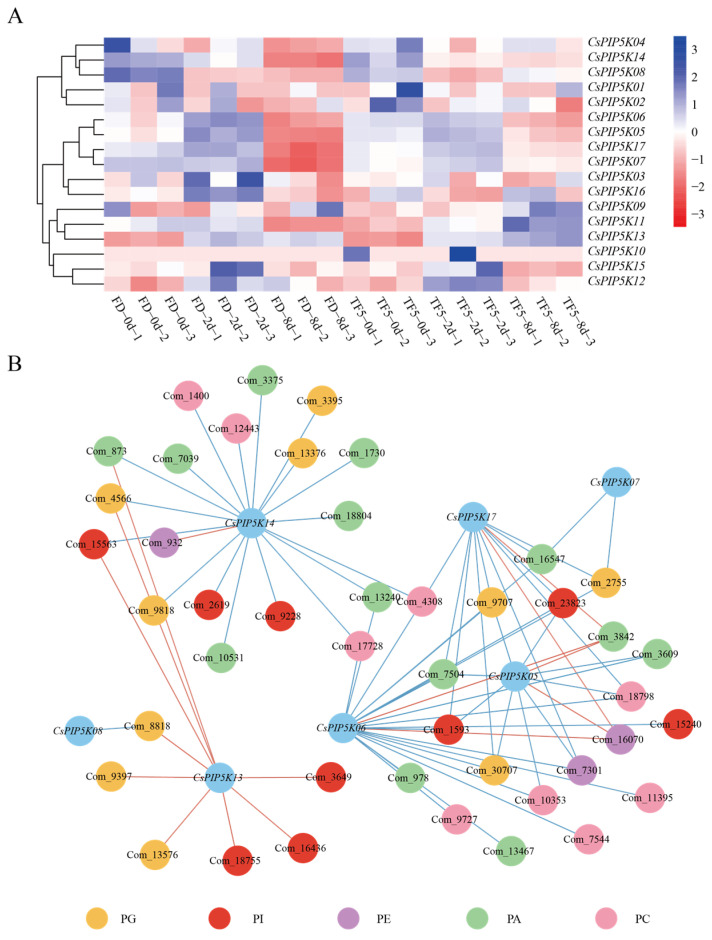
Transcriptome and metabolome analysis of the *CsPIP5K*s under spider mite infection. (**A**) Expression patterns of *CsPIP5K*s in different cultivars at various infestation time points; (**B**) Correlation analysis between *CsPIP5K*s and compounds related to phospholipid metabolism. Red line indicates positive correlation, blue line indicates negative correlation, and the darker the color, the stronger the correlation.

**Figure 9 genes-15-00932-f009:**
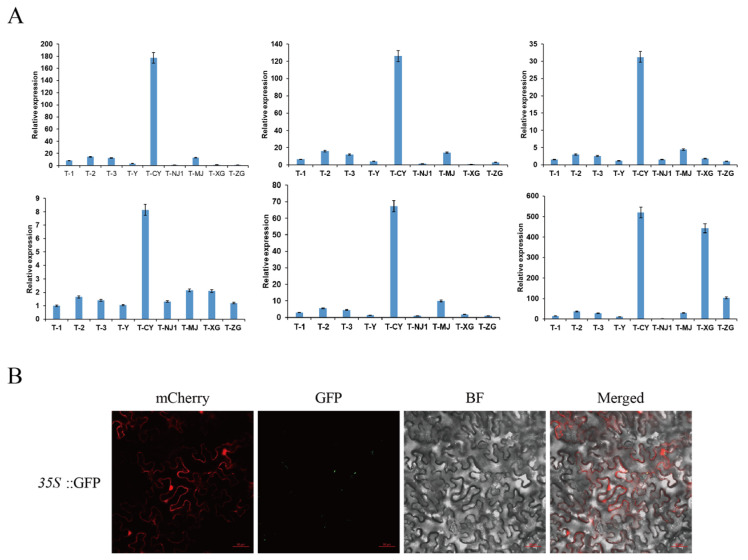
(**A**) Results of qPCR experiments of *CsPIP5K06* in Tianfu-5 multi tissue; (**B**) results of subcellular localization of *CsPIP5K06*. GFP is short for Green Fluorescent Protein, and BF is short for Bright Field.

## Data Availability

The original contributions presented in the study are included in the article, further inquiries can be directed to the corresponding author.

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
