# Peer review of "Genome-Wide Identification of the *PIP5K* Gene Family in *Camellia sinensis* and Their Roles in Metabolic Regulation"

_genes, 2024, doi:10.3390/genes15070932_

Round 1
Reviewer 1 Report
Comments and Suggestions for Authors
The manuscript in reference describes the identification of PIP5K gene family from Camelia sinensis presumably under spider mite infestations. The manuscript has interesting elements but requires revisions.
1. The title seems to be incomplete, i.e., the final is missing, possibly "spider mite infestation or presence or pressure"?
2. Regarding the previous comment, althougt spider mite is mentioned in title and other passages of the manuscript, the effect of spider mite infestations was not developed in the manuscript, so it can not be included as a context since the maniscript is only dedicated to the characterization and identification of the PIP5K gene family from C. sinensis. In this regard, as no experiment was conducted with spider mite-infested plants, I recommend to clarify these passages from this study to avoid wrong intrepretations by readers.
3. Line 14: The relationship between metabolomics and transcriptomics is not clear since the metabolomes were not analyzed from the same accessions that were analyzed in this study. Be consistent throughout the manuscript.
4. Lines 20 and 85: The induction by spider mites is not clear since no experiments with spider mite-infested plants were conducted in this manuscript and a previous reported dataset was used. This information must be clarified to be used as an appropriate justification of the study. Be consistent throughout the manuscript.
5. Line 37: Does gene catalyze or the enzymes encoded by the genes? Revise this sentence and throughout the manuscript.
6. What type of data of reported metabolites was used to calcuate the Pearson coefficients? concentrations? abundance? semi-quantitative levels based on intensity? This information must be clarified since it is crucial to demonstrate the accuracy of these correlations between previous reported data.
7. Line 149: The information of the parent plants used to extract RNA should be completed since the type and status of tissues must be provided.
8. Discussion must be improved since the line is challenging to be followed.
9. Conclusions are very general and laconically organized. A more specific conclusions from the conceptual findings must be provided.
Comments on the Quality of English Language
A deep scrutiny of the manuscript looking for some grammar and stylistic issues must be performed to improve readability and clarity of several passages.
Author Response
感谢您指出问题并提供您的建议,我们认为这些建议非常重要。您的宝贵建议准确地指出了我们手稿中的不足之处和可能使读者感到困惑的部分。我们根据您的建议对稿件进行了一一修改,并回答了您的问题。再次感谢您审阅我们的稿件。
- 标题似乎不完整,即最终缺失,可能是“蜘蛛螨侵扰或存在或压力”?
R:这是一个很好的建议。结合以下评论,我们也认为我们在整篇文章中没有进行感染实验。因此,为了避免读者的混淆,我们将标题改为“山茶花PIP5K基因家族的全基因组鉴定及其在代谢调控中的作用”
- 关于前面的评论,在手稿的标题和其他段落中提到了 althougt 蜘蛛螨,手稿中没有开发蜘蛛螨侵扰的影响,因此不能将其作为上下文包括在内,因为 maniscript 仅致力于表征和鉴定来自 C. sinensis 的 PIP5K 基因家族。在这方面,由于没有对蜘蛛螨侵扰的植物进行实验,我建议澄清本研究中的这些段落,以避免读者误入歧途。
记者:非常感谢你的建议。我们还发现,我们的手稿可能会因为过多地提到蜘蛛螨而使读者感到困惑。因此,我们在整篇文章中删除了对蜘蛛螨的不必要描述。例如,在之前的评论中,我们修改了标题并删除了“蜘蛛螨”。为了强调我们使用的是以前报道的数据,而不是蜘蛛螨的实验,我们更改了文章中几个段落的描述。例如,第17-19行改为“总体而言,本研究不仅全面阐明了CsPIP5K基因家族的特征、结构和进化,还根据先前发表的数据鉴定了几个与磷脂生物合成相关且与红蜘蛛相关的候选CsPIP5K基因”
- 第 14 行:代谢组学和转录组学之间的关系尚不清楚,因为代谢组不是从本研究中分析的相同种质中分析的。在整个手稿中保持一致。
R:很抱歉我可能造成了混乱,但我们并不完全同意你的观点。虽然这项研究没有产生代谢组学和转录组学数据,但我们根据先前从实验室收集的数据探索了 CsPIP5K 的代谢调控。为了澄清这一点,我将这句话修改为:“为了充分了解蜘蛛螨感染下CsPIP5K的代谢反应,我们整合了以前发表的代谢组学和转录组学数据”。
- 第 20 行和第 85 行:蜘蛛螨的诱导尚不清楚,因为本手稿中没有对蜘蛛螨感染的植物进行实验,并且使用了先前报道的数据集。必须澄清这些信息,以便用作研究的适当理由。在整个手稿中保持一致。
R:非常感谢你提出这个问题,我们认为这个问题与前面的问题几乎相似。我们在全文中标明了所得结果的数据来源,以免读者认为我们在本研究中进行了数据生产。修改了类似表述:将CsPIP5K由蜘蛛螨诱导的基因家族改为可能与蜘蛛螨侵扰有关的结论。如第85行中的“最后,我们挖掘出一个名为CsPIP5K06的基因,该基因具有调节多种脂质的可能,并与蜘蛛螨感染有关”。
- 第 37 行:基因催化或基因编码的酶吗?修改这句话和整个手稿。
R:应该是基因编码的酶,我们在第 37 行进行了更正,并检查了整个文本是否有修改。
- 使用什么类型的报告代谢物数据来计算皮尔逊系数?浓度?丰富?基于强度的半定量水平?必须澄清这些信息,因为证明以前报告的数据之间的这些相关性的准确性至关重要。
记者:谢谢你的评论。我们使用的代谢组学数据是以前在我们的实验室发表的研究,我们使用代谢物的丰度计算了相关性。我们更改了材料方法部分的描述,“基于我们实验室先前发表的不同时间点感染红蜘蛛螨的不同茶品种的代谢谱数据,结合相应的转录组学谱,我们利用 R 语言中的 cor.test() 函数计算了转录水平和 CsPIP5K 基因家族成员代谢物丰度之间的 Pearson 相关系数”第 141-145 行。
- 第 149 行:由于必须提供组织的类型和状态,因此应填写用于提取 RNA 的亲本植物的信息。
R:RNA提取样本取自四川省雅安市明山区中丰乡海棠村牛聂平基地正常生长的2年生天府5号。我们收集了芽,第一片叶子,第二片叶子,第三片叶子,成熟叶子,嫩茎,老茎,主根和须根。样品状态:(TF5-Y)芽,(TF5-1)第一叶,(TF5-2)第二叶,(TF5-3)第三叶,(TF5-CY)成熟叶,(TF5-NJ)嫩茎,(TF5-LJ)老茎,(TF5-ZG)主根,(TF5-XG)须根。它们同时取自同一株植物。每个样品分别从 5 株植物取样后混合。
- 必须改进讨论,因为这条路线很难遵循。
记者:非常感谢你的建议。我们也同意你的观点,即讨论部分太长,过渡太突然。因此,我们花了很多精力来修改它。在新稿中,我们主要讨论了三个问题。首先,我们将这项研究的多个结果与其他物种的研究进行了比较,突出了这项研究的独特之处以及多个物种之间的共同点。其次,我们探讨了PIP5K防御机制背后的复杂代谢调控网络。最后,我们讨论了本研究中发现的关键候选基因CsPIP5K06的潜在调控机制。因此,我们重新组织了整个讨论部分,并重写了大部分句子,以确保内容连贯和合乎逻辑。
- 结论非常笼统且组织简洁。必须从概念调查结果中得出更具体的结论。
记者:非常感谢你提出这个重要问题。事实上,我们在写这部分时有很多不足之处。现在,在修订后的手稿中,我们突出了我们的发现,并在第 456-463 行中用简洁的句子总结了我们的重要结果。

Reviewer 2 Report
Comments and Suggestions for Authors
Majors:
- authors studied effects of spider mite infection on phospholipid metabolism in tea, but nowhere say which spider mite was used (species name) in the experiments and how infection was done
- numbering of figures and tables, also in the supplementary material, is wrong. Table S1, for example does not show the primers used.
- why does Fig. 6 show only 15 of the 17 CsPIP5K genes and why only 5 out of 11 for Arabidopsis?
- I would like to see a better separation between description of results and interpretation of the data (Discussion). For example, lines 371 -375 belong to the Discussion.
qPCR results in Fig. 9 must show a statistical evaluation of the data.
Minors:
- line 111: species names of tomato and grape must be given when first mentioned
- line 151 and others: use consistent abbreviations (either s or sec, etc.)
- line 173 and others: give all species names in italics, also in the References
- line 190: is it necessary to give molecular weights with two decimals?
- legend to Fig. 3: B and C are not indicated in the figure
- line 304 and others: explain the acronyms when used at first
- line 386: do not begin a sentence with "and"
- Ref. 4, 26 and 37 are not complete. Give titles of the papers either in uppercase or in lowercase letters, but not mixed.
Comments on the Quality of English Language
English needs minor improvement.
Author Response
Thank you so much for your insightful comments and invaluable suggestions. They are truly significant to us. Your astute observations have pinpointed the areas in our manuscript that needed improvement and those that might confuse our readers. We have carefully revised the manuscript based on your suggestions and responded to your questions. Once again, we deeply appreciate your time and effort in reviewing our manuscript. Your feedback is truly appreciated!
- authors studied effects of spider mite infection on phospholipid metabolism in tea, but nowhere say which spider mite was used (species name) in the experiments and how infection was done
R: We are very sorry that we did not give detailed information. We supplemented the materials and methods. We used six-spotted spider mite (Eotetranychus sexmaculatus). The insect was collected from the organic tea garden base of Zhougongshan Tea industry in Yucheng District, Ya 'an, in mid-April 2021, and was bred by Fudingdabai's new shoot 1 bud 5 and 6 leaf branches for a long time. The feeding conditions were (26±1) ℃ and relative humidity was 80%.
- numbering of figures and tables, also in the supplementary material, is wrong. Table S1, for example does not show the primers used.
We are very sorry that this is indeed our mistake. The primers used should be listed in Table S6. We have checked the manuscript and corrected such mistakes.
- why does Fig. 6 show only 15 of the 17 CsPIP5K genes and why only 5 out of 11 for Arabidopsis?
I'm sorry, but we don't quite understand your confusion. We show all collinear relationships in Figure 6 and list specific genes in Table S3. We show 11 Arabidopsis and tea gene pairs in Figure 6, not just 5.
- I would like to see a better separation between description of results and interpretation of the data (Discussion). For example, lines 371 -375 belong to the Discussion.
qPCR results in Fig. 9 must show a statistical evaluation of the data.
R: We very much agree with your point of view, in the revised manuscript we have moved the contents of the original Line 371-375 to the discussion section of Line 447-452. As for the second question, we took three biological replicates to conduct experiments, and we also provided the error bar of the three experiments in Figure 9. But as you mentioned, we omitted a statistical evaluation of the data, so we added this to the corresponding description. “From the results, the three biological replicates all have small error bar, indicating that the experimental results have good accuracy and consistency” in line 377-379.
Minors:
- line 111: species names of tomato and grape must be given when first mentioned
We are very sorry that this is our mistake, we should have shown the Latin name when it first appeared. In the revised manuscript, we have made changes in the corresponding positions and marked them with red font.
- line 151 and others: use consistent abbreviations (either s or sec, etc.)
R: Thank you very much for your careful discovery. We have also noticed that the abbreviations before and after the article are not consistent. Therefore, we checked line by line and corrected and supplemented, for example, line151 was changed to " Total RNA was extracted from multiple tissues of “Tianfu-5” (a widely studied tea variety, abbreviated as TF5) using Vazyme (RC411) Kit according to the manufacturer’s specifications".
- line 173 and others: give all species names in italics, also in the References
Thank you very much for your reminding, we have checked the whole article and fixed the problem that the species name should be italicized.
- line 190: is it necessary to give molecular weights with two decimals?
R: Thank you for your reminding, but we do not think it is necessary. We have modified line 190 and checked the same problem in the whole text.
- legend to Fig. 3: B and C are not indicated in the figure
R: We are very sorry for such a mistake. The reason for this issue was that our figure captions caused confusion. We have revised the captions accordingly.
- line 304 and others: explain the acronyms when used at first
R: This is an important question, and the lack of a full name can really confuse readers. We have reviewed the full text, adding the full name of the acronym that first appeared.
- line 386: do not begin a sentence with "and"
R: Well, we have made modifications in the corresponding lines and checked for similar issues.
- Ref. 4, 26 and 37 are not complete. Give titles of the papers either in uppercase or in lowercase letters, but not mixed.
R: Thank you for raising the issue. It was our carelessness in formatting. We have now corrected all similar problems in the references and highlighted them in red.
